# TLR7 and IgM: Dangerous Partners in Autoimmunity

**DOI:** 10.3390/antib12010004

**Published:** 2023-01-06

**Authors:** Timm Amendt, Philipp Yu

**Affiliations:** 1Institute of Immunology, University Hospital Ulm, Ulm University, 89081 Ulm, Germany; 2The Francis Crick Institute, London NW1 1AT, UK; 3Institute of Immunology, Philipps-Universität Marburg, 35037 Marburg, Germany

**Keywords:** autoantibodies, IgM, B cells, autoimmunity, TLR7

## Abstract

The B cell antigen receptor (BCR)-repertoire is capable of recognizing a nearly unlimited number of antigens. Inevitably, the random nature of antibody gene segment rearrangement, needed in order to provide mature B cells, will generate autoreactive specificities. Once tolerance mechanisms fail to block the activation and differentiation of autoreactive B cells, harmful autoantibodies may get secreted establishing autoimmune diseases. Besides the hallmark of autoimmunity, namely IgG autoantibodies, IgM autoantibodies are also found in many autoimmune diseases. In addition to pathogenic functions of secreted IgM the IgM-BCR expressing B cell might be the initial check-point where, in conjunction with innate receptor signals, B cell mediated autoimmunity starts it fateful course. Recently, pentameric IgM autoantibodies have been shown to contribute significantly to the pathogenesis of various autoimmune diseases, such as rheumatoid arthritis (RA), autoimmune hemolytic anemia (AIHA), pemphigus or autoimmune neuropathy. Further, recent studies suggest differences in the recognition of autoantigen by IgG and IgM autoantibodies, or propose a central role of anti-ACE2-IgM autoantibodies in severe COVID-19. However, exact mechanisms still remain to be uncovered in detail. This article focuses on summarizing recent findings regarding the importance of autoreactive IgM in establishing autoimmune diseases.

## 1. Introduction

Maintaining physiological homeostasis is crucially dependent on a well-balanced immune system. Integral to that balance is the avoidance of reacting to self-structures, so-called autoimmune reactions [1,2]. B cells are fundamental players in humoral immunity by producing antibodies capable of recognizing a nearly unlimited number of antigens [3]. However, the random nature of providing this enormous B cell antigen receptor (BCR), and thus antibody-repertoire, inevitably leads to the production of self-reactive B cells [3,4]. In order to provide an extremely divers BCR repertoire, early B cell progenitors undergo antibody gene rearrangement (V—variable, D—diversity, J—joining) in the bone marrow [3]. Autoreactive clones recognizing self-structures with a high autoantigen-affinity undergo either clonal deletion or receptor editing (central tolerance) [5,6,7]. Nevertheless, autoreactive B cells that circumvent this mechanism and home to the spleen may be subjected to peripheral tolerance mechanisms, such as the downmodulation of IgM-class BCR expression and maintenance of IgD-class expression, rendering these B cells unresponsive to monovalent autoantigens [8,9,10,11,12]. It has been demonstrated that autoreactive IgD^+^ B cells can get activated when encountering autoantigens in a polyvalent form (complex) [10,11,12]. Physiologically, B cells co-express IgM- and IgD-class BCRs and cognate antigen induces a combination of IgM and IgD-mediated signaling but exact mechanisms are yet to be uncovered in detail [13].

Most mature B cells have been demonstrated to require the engagement of autoantigen with their BCR in order to reach the periphery [14,15]. Further, we know an enormous number of autoantibody-mediated autoimmune diseases with a drastically increasing prevalence [1,2,16,17]. In particular, autoantibodies are capable of targeting organ-specific receptors, such as the acetyl choline receptor (AchR), leading to myasthenia gravis [2], or induce immune complex-mediated systemic inflammation, such as anti-nuclear-antibodies (ANA), as in systemic lupus erythematosus (SLE) [18,19].

Interestingly, most diagnosis criteria focus on the presence of IgG autoantibodies ignoring autoreactive IgM [1,18,20,21]. In contrast, patients that are incapable of class-switching to IgG, thus only able to produce IgM antibodies (hyper-IgM-syndrome [HIGM]) often show severe autoimmunity [22,23]. Here, autoreactive IgM targeting red blood cells (autoimmune hemolytic anemia (AIHA)) or thrombocytes (idiopathic thrombocytopenia purpura (ITP)) lead to a drastic autoimmune phenotype [24,25]. This underlines the importance of autoreactive IgM in autoimmune pathogenesis.

IgM antibodies either occur in a pentameric state containing a J-chain, or hexameric state not bearing a J-chain and being of very short half-life [26,27]. Since most of the circulating IgM antibodies do not harbor antigen-affinity increasing mutations by somatic hypermutation (SHM), the polymeric form compensates a low antigen-affinity by avidity [26,28]. Further, IgM antibodies are highly potent inflammation drivers due to the ability of recruiting complement effectively [29,30,31] (Figure 1). In addition, IgM antibodies are heavily glycosylated bearing 5 N-linked glycosylation sites with complex or high-mannose type glycans [32]. In comparison, IgG antibodies only bear one N-linked glycosylation site at the CH2 domain. Recently, it was shown that the type of glycans being present is important for the effector function, as anti-T cell IgM severely blocks T cell proliferation and T cell responses solely when being sialylated [32,33].

Taken together, the abundance of autoreactive B cells and unique structure and functions of IgM antibodies indicate a crucial role in the pathogenesis of autoimmune diseases. The old IgG autoantibody-centric concept is advancing towards an IgM autoantibody-including concept considering IgM’s capability of establishing autoimmunity. In this article, we will discuss recent advances highlighting the role of IgM autoantibodies in autoimmunity.

## 2. Natural IgM: Self-Reactive House-Cleaning

Autoreactive antibodies that are not directly referred to as harmful autoantibodies causing autoimmune diseases, are called natural autoantibodies [34]. The most prominent class of natural autoantibodies is represented by natural IgM (nIgM) accounting for the vast majority of serum IgM [35]. These nIgM antibodies are characterized by a low antigen-binding affinity as well as strong polyreactivity [36,37,38]. Interestingly, nIgM antibodies are produced independent of antigen encounter and target mostly harmful self-molecules such as advanced glycation end-products (AGE) or oxidized lipids (oxLDL) in order to limit autoinflammation [30,35,39,40]. This function is important to remove apoptotic cells and harmful protein aggregates as well as DAMPs (danger-associated molecular patterns) [30,35,39,40]. Further, nIgM antibodies are encoded by limited germline V(D)J sequences in the genome of B1 and marginal zone (MZ) B cells [30,39,41,42]. As mentioned, limitations of the restricted repertoire of nIgM antibodies are overcome by strong polyreactivity [34]. In contrast, primary IgM occurring in early phases of infections is thought to be produced by canonical B2 B cells [26].

Several elegant studies have shown that mice lacking nIgM are highly susceptible to atherosclerosis, lupus-like syndromes and autoinflammation [30,39,41,42]. In agreement with mouse data [30], patients suffering from selective IgM deficiency (sIgMD) are prone to develop autoimmune diseases highlighting the importance of nIgM in homeostasis [43]. In addition, decreased numbers of MZ B cells being the major source of IgM secreting cells, have been demonstrated to increase the risk for autoimmune diseases most likely by lowering nIgM titers thus highlighting their importance for homeostasis [44]. Recently, in a clinically relevant context, it has been demonstrated that nIgM is capable of preventing thrombosis [45] showing the great therapeutic potential of nIgM.

In summary, the absence or dysfunction of nIgM has been demonstrated to be associated with autoimmune diseases. Importantly, nIgM is not inducible by autoantigens such as autoreactive primary IgM is, which is specifically targeting self-structures. In contrast to nIgM, the presence of autoreactive primary IgM (hereafter referred to as *autoreactive IgM*) is associated with autoimmune diseases.

## 3. COVID-19: Autoreactive IgM as Indicator for Disease Severity

The SARS-CoV-2 virus emerging in 2019 that lead to a world-wide pandemic situation can cause a severe virus-mediated pneumonia (COVID-19) [46]. Further, SARS-CoV-2-triggered autoimmunity has been intensely debated [47,48]. Several studies found a vast variety of autoreactive IgG antibodies including ANA-IgG, anti-Ro-IgG or anti-cardiolipin IgG [49,50]. However, most studies focused on the examination of autoreactive IgG antibodies investigating a potential link of autoantibodies to diseases.

Interestingly, several recent studies linked autoreactive IgM antibodies recognizing prothrombin [51] or ACE2 [52] directly to COVID-19 disease severity. Emmenegger and colleagues found that IgM autoantibodies, but not IgG autoantibodies, are associated to SARS-CoV-2 infection and SARS-CoV-2 antibody response [51]. The study reports IgM autoantibodies exclusively targeting phosphatidic acid (PA), phosphatidylserine (PS), annexin V (AnV), β2 glycoprotein I (β2) and prothrombin (PT). In order to clear damaged cells, including phospholipid motifs, nIgM is required in sufficient titers. However, since uninfected subjects did not show anti-phospholipid IgM when compared to SARS-CoV-2 infected individuals, the involvement of nIgM is unlikely. Further, anti-phospholipid IgM titers correlating with anti-SARS-CoV-2 antibody titers point at infection induced autoreactive primary IgM. Interestingly, anti-prothrombin IgM autoantibodies of SARS-CoV-2 infected individuals do not recognize spike protein since both target proteins share very limited structural similarities. Thus, cross-reactivity and molecular mimicry can be excluded, supporting the notion of primary autoreactive IgM. However, anti-PT IgM should still be able to neutralize PT and thus interfere with blood coagulation contributing to disease severity.

In summary, the production of autoreactive IgM upon infection that does almost certainly not belong to the class of nIgM, indicates the presence of an autoreactive canonical B cell population capable of differentiating into IgM^+^ plasma cells.

However, the exact mechanism of anti-PT IgM generation in SARS-CoV-2 infected human individuals needs to be uncovered.

## 4. TLR7 as an Exemplary Innate Pattern Recognition Receptor for Autoimmune or Anti-RNA Virus Specific IgM^+^ B Cell Activation

The unique importance of the IgM BCR signal in triggering an antigen-specific response or B-cell lymphoproliferative disorders is well established. In contrast, to what extent other signal pathways contribute to drive autoreactive IgM^+^ B cells is still not completely understood. Mounting evidence suggests that in addition to the IgM-BCR itself, cytokine receptors, receptors mediating cognate T cell-B cell interaction, and last but not least, innate immune receptors provide the constellation of signals that allow the activation, survival and expansion of autoreactive B cells [53].

As an example for B cell intrinsic innate immune receptors we will focus on the role of Toll-like receptor 7 (TLR7). It belongs to the sub-family of innate pattern recognition receptors that sense nucleic acids. TLR7 has drawn considerable interest because its ligand (guanosine and ssRNA), its structure and molecular activation mechanisms are relatively well understood [54]. Recent findings suggest that differences in subcellular sorting by Unc93b1 and syntenin-1 differentiate an autoimmunity promoting function of TLR7 from the protective effects of TLR9 in murine models [55,56]. New venues of research to understand its role in autoimmunity have opened by murine models with loss of function and upregulated TLR7 activity like the yaa model (a chromosomal translocation that contains the murine TLR7 gene) [57] and most importantly the characterization of a human TLR7 gain-of-function mutation in a lupus patient and in a mouse strain with the identical mutation [58]. The patient with lupus symptoms confirms the direct link between TLR7 and human B cell autoimmunity. The promotion of a lupus phenotype by TLR7 is supported by the fact that a mouse strain with the yaa translocation and therefore an additional copy of TLR7 developed accelerated symptoms of lupus pathology and anti-DNA autoantibodies. Consistently, the genetic ablation of TLR7 abolished the development of spontaneous germinal centers in murine lupus models coinciding with amelioration of pathogenesis in these models [59]. The new data on the gain-of-function mutation in the human patient is important because it links pathogenesis with a B cell intrinsic effect that results in activation of autoreactive B cells in human and mouse. In particular an experiment where purified B cells (from the gain-of-function TLR7^Y264H^ mice) treated with BCR crosslinking anti-IgM in vitro react with enhanced survival of B cells is interesting. It suggests that mutant TLR7 signaling without the addition of TLR7 ligand is directly enhancing IgM-mediated BCR signaling. The fact that anti-IgM promotes the survival and induction of genes linked to survival of B cells suggests that in this patient and the homologous murine model the threshold to autoreactivity in B cells might be overcome by IgM-BCR signals in conjunction with the enhanced TLR7 signal.

The interplay between BCR and TLR7 signaling is complex and needs more research efforts to further our understanding. For example, genetic BCR ablation impairs the capacity of B cells to proliferate to TLR4 [60,61]; whether TLR7 ligand driven proliferation is equally affected is unknown. Although a role of a MyD88-independent signaling pathway for TLR4 via SYK-AKT-ERK has been established, no information exists if TLR7, which is believed to exclusively signal through MyD88, is actually behaving similarly.

The other way round, namely the influence of TLR7 towards the BCR signal, is even less well understood. Interestingly, the TLR7^Y264H^ mutant results in increased IgM-mediated proliferation in vitro, without TLR7 ligand engagement [58]. It is possible that a higher sensitivity of mutant TLR7 to recognize guanosine or endogenous RNA species might result in an elevated cooperative signal with IgM.

With regard to the role of IgM^+^ B cell response and TLR7 in SARS-CoV-2 infection, new aspects came recently into focus. The question needs to be addressed whether autoreactive IgM is part of the pathology of COVID-19 or long covid symptoms, which affect between 5.8 (vaccinated) and 7% (not-vaccinated) of infected patients. Surprisingly, genetic analysis of young brothers who developed severe COVID-19 identified for the first-time humans with TLR7-deficiency [62]. At present, it is not clear whether the TLR7 loss of function results in a pleiotropic immunodeficiency or is dominated by a B cell intrinsic effect of TLR7. However, various RNA-virus models of TLR7-deficient mice, including murine endogenous retrovirus, demonstrated that TLR7 is absolute essential to mount a RNA-virus specific B cell response [63]. It would be interesting if the severe COVID-19 outcome of TLR7-deficient human patients is due to an impaired anti-SARS-CoV-2 B cell response.

## 5. Autoreactive IgM: First Step in Establishing Autoimmune Diseases?

For decades, diagnosing autoantibody-borne autoimmune diseases was focused on detecting certain autoreactive IgGs (see above) [1,2]. However, several recent studies supported a drastic shift in the view of the involvement of IgM autoantibodies in autoimmune diseases.

An example is the current view of the pathogenesis of bullous pemphigoid disease, which was thought to be exclusively restricted to IgG autoantibodies [64]. In brief, bullous pemphigoid is characterized by skin lesions and itchy bullae caused by autoantibodies targeting components of hemidesmosomes [64]. However, Hirano et al. recently showed that bullous pemphigoid can be exclusively caused by autoreactive IgM [65]. By using super-resolution imaging, the authors detected BP180 as the cognate autoantigen of the reported IgM autoantibodies. Further, histological analyses confirmed the notion of anti-BP180 IgM being involved in bullae formation. Thus, anti-BP180 IgM is capable of binding its autoantigen with sufficient affinity and avidity sterically hindering hemidesmosome formation. This fact opens up speculations about the origin of anti-BP180 IgM that possibly underwent affinity maturation during germinal center reaction, or at extrafollicular sites [66,67,68,69,70]. Further, the efficient recruitment of complement by IgM could further promote skin inflammation more drastically than IgG autoantibodies. Consequently, the authors suggest a new sub-class of pemphigoid disease, namely IgM pemphigoid, further supporting a shifting of the view towards autoreactive IgM causative for autoimmune diseases.

Another autoimmune disease where autoreactive IgG is a hallmark for diagnosis is resembled by rheumatoid arthritis (RA) [71]. Here, IgG autoantibodies binding to citrullinated (ACPA), acetylated or carbamylated self-antigens are classified as anti-modified protein antibodies (AMPA) [72]. Whether AMPA IgG require excessive somatic hypermutation (SHM) in order to bind modified proteins in a cross-reactive manner has been debated intensely. Interestingly, Reijm et al. [73] have suggested that AMPA IgG might originate from AMPA IgM possessing germline configuration. The authors further show that AMPA IgM detected in RA patients can readily bind modified self-proteins and thus induce pathology. In addition, when expressing AMPA IgM as AMPA IgG, modified self-protein binding was lost, indicating a crucial role of the pentameric structure of AMPA IgM. Together, these results imply that AMPA IgM occurs early in RA development leading to joint damage, thus establishing the disease whilst further autoreactive B cells switch to IgG and perform SHM and eventually secrete AMPA IgG.

The concept of IgM autoantibodies as pathogenic and causative drivers of autoimmune diseases is further supported by a recent study reporting on therapeutic plasma exchange in patients suffering from autoimmune neuropathy [74]. In this study, a group of small fiber neuropathy (SNF) patients associated with trisulfated heparan disaccharide (TS-HDS) IgM autoantibodies were treated by plasma exchange. Strikingly, the depletion of TS-HDS IgM autoantibodies in these patients lead to a reduction of symptoms such as lower extremity parasthesia by over 70%. Thus, the results of Olsen et al. [74] indicate that TS-HDS IgM is responsible for symptoms observed in SNF.

Furthermore, Kawakami and colleagues [75] recently tested if anti-phosphatidyl-serine/prothrombin complex (PS/PT) IgM autoantibodies are capable of causing cutaneous ulcers in patients with cutaneous vasculitis. The authors demonstrated that rats injected with anti-PS/PT IgM developed cutaneous vasculitis showing that anti-PS/PT IgM autoantibodies as direct cause for disease.

In addition to studies showing autoantigen-specific IgM causative for pathology in certain autoimmune diseases, patients suffering from HIGM are also prone to developing autoimmune diseases. In particular, HIGM patients only possess polyreactive IgM being incapable of removing potential autoreactivity by SHM. For instance, recent studies have shown that HIGM patients can develop primary biliary cholangitis (PBC) caused by anti-MIT3-IgM autoantibodies not present in healthy subjects [76,77].

In sum, several recent studies showing that autoreactive IgM can be the first step in establishing autoimmune diseases highlight the opportunity of early screening by using autoreactive IgM as a diagnostic marker. Consequently, testing patients with mild symptoms in early disease phases for autoreactive IgM, that would normally not receive treatment due to the lack of IgG autoantibodies, could open up a therapeutic intervention window.

## 6. Conclusions and Outlook

The beneficial effects of nIgM targeting harmful and altered self-molecules, such as oxidized lipids, have been studied extensively [35]. For instance, nIgM has been shown to protect from thrombosis or atherosclerosis [41,45]. Importantly, nIgM is produced antigen encounter-independently and possesses a strong polyreactivity being capable of binding dsDNA [30,36].

However, in healthy individuals the nIgM repertoire is devoid of specificities that specifically target cell surface receptors. Autoreactive IgM antibodies that bind for instance BP-180 as in pemphigus [65], are referred to as acquired autoreactive IgM and do not belong to the nIgM compartment. This class of antibodies binds autoantigens and consequently affects its functions or leads to cell elimination [26]. Several recent studies have demonstrated that autoreactive IgM can be causative for autoimmune symptoms as anti-BP180 IgM leads to IgM pemphigoid [65], anti-PT IgM antibodies interfere with blood coagulation in severe COVID-19 patients [51,78], or AMPA-IgM as disease driver in RA [73]. In conclusion, these studies and others indicate that autoreactive IgM is capable of causing autoimmune diseases in the absence of IgG autoantibodies (Figure 2). Thus, assessing IgM autoantibody titers in patients presenting with autoimmune symptoms might help to find an adequate treatment faster.

If the hypothesis holds water that IgM^+^ B cells are involved in the primary step to autoimmunity, and their activation is dependent on TLR7 ligands conveying an aberrant signal to these cells, pharmacological suppression could be beneficial for the plethora of IgM mediated autoimmune diseases discussed. At present, at least three independent research groups came up with antagonistic/inhibitory small molecules and a TLR7 specific monoclonal antibody to suppress TLR7 activation [79,80,81]. It would be most informative to analyze IgM autoantigen levels in autoimmune models and patients undergoing treatment with this TLR7 inhibiting reagents.

Since it has been shown that TLR7 gain-of-function mutations [58] lead to lupus-like diseases in humans and mice, it is conceivable that TLR7 together with BCR signaling initiate the activation of autoreactive B cells and differentiation into short-lived IgM^+^ autoreactive plasma cells (SLPC). In early disease phase, the rapid and strong activation of autoreactive B cells and the possible absence of T cell help prior to GC-entry [82,83], might result in a massive production of autoreactive IgM. Depending on the autoantigen targeted by autoreactive IgM, these autoantibodies could act by different effector mechanisms.

The mechanisms by which IgM autoantibodies induce pathology might differ from effects that IgG autoantibodies with identical specificities exert on targets. First, the pentameric structure of IgM and the lack of Fab flexibility, compared to the highly flexible hinge region of IgG [84], could alter the mobility of cell surface receptors bound by antibodies. For instance, IgM autoantibodies binding to components of tight junctions as in pemphigus, could block cell-cell interactions more efficiently than IgG autoantibodies due to its relative stiffness.

Second, IgM antibodies are efficient in forming immune complexes due to their pentameric structure and often polyreactivity. This feature might be particularly relevant when targeting soluble autoantigens, such as PT or PA, as described in COVID-19 patients [51]. In particular, anti-PT/PA IgM autoantibodies might be capable of binding high numbers of self-molecules at once, form immune complexes efficiently and deplete the self-molecules from circulation via uptake by macrophages [85]. Thus, COVID-19 patients could lose enormous numbers of PT molecules rapidly leading to severe effects on blood coagulation.

Third, IgM has been shown to recruit complement more efficiently compared to IgG [26,86]. Thus, complement-mediated tissue damage might be enhanced when IgM is the isotype of autoantibody present.

Since the effects of IgM autoantibodies are yet poorly understood, several questions besides disease-specific mechanisms arise:(1)Do all antibody-mediated autoimmune diseases that get diagnosed by the presence of IgG autoantibodies show early phases of IgM autoantibodies?(2)Do IgM autoantibodies showing identical specificity as IgG autoantibodies induce different symptoms?(3)How are IgM^+^ B cells allowed to differentiate into IgM-secreting plasma cells?(4)What is the role of TLR7 in the activation of autoreactive B cells eventually secreting IgM autoantibodies?

In sum, there is a growing body of evidence suggesting that TLR7 is capable of controlling the activation of autoreactive IgM^+^ B cells. Further, autoreactive B cells that differentiate into IgM autoantibody-secreting plasma cells might cause a vast variety of autoimmune diseases via pathogenic IgM autoantibodies.

## Figures and Tables

**Figure 1 antibodies-12-00004-f001:**
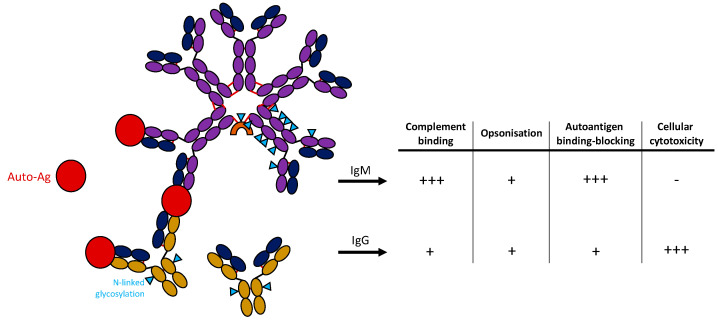
Structure of IgM and IgG antibodies and potential function in autoimmunity. IgM antibody is shown in a pentameric form containing a J-chain (orange) binding autoantigen (red). IgG monomers are shown below. N-linked glycosylation sites of antibodies are marked with a blue triangle. Table on the right shows critical features of antibodies comparing IgM and IgG antibodies.

**Figure 2 antibodies-12-00004-f002:**
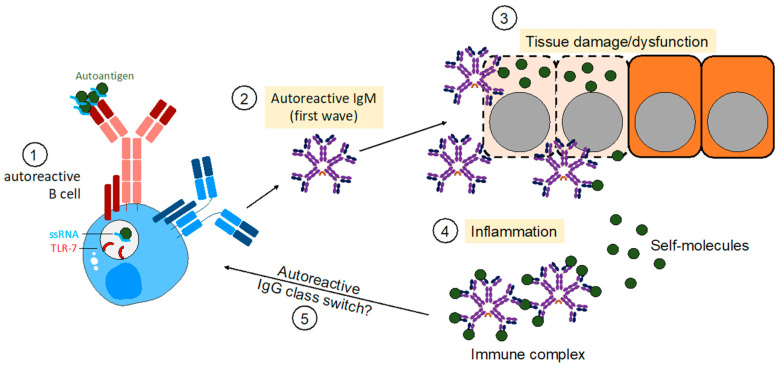
Autoreactive IgM in establishing autoimmunity. Model of how initial TLR7 signals induce autoreactive B cell activation of self-reactive B cells. In a subsequent first wave, autoreactive B cells produce IgM autoantibodies binding to self-antigens thereby inducing neutralization of targets or tissue damage. Consequently, tissue damage establishes an inflammatory environment leading to the activation of further self-reactive B cells. In the end, autoreactive IgM is capable of causing autoimmune diseases without the presence of IgG autoantibodies.

## Data Availability

No new data were created or analyzed in this study. Data sharing is not applicable to this article.

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
