# Peer review of "TLR7 and IgM: Dangerous Partners in Autoimmunity"

_2073-4468, 2023, doi:10.3390/antib12010004_

Round 1
Reviewer 1 Report
The subject of the connection of secreted IgM to autoimmunity is timely, and certainly of importance. This review is well written, thought provoking, and includes a current and comprehensive reference list. The discussion of the connection of COVID 19 to IgM dependent autoimmune phenomena is especially well timed.
What is lacking though is some discussion of marginal zone (MZ) B cells, which would bring the whole discussion of secreted IgM into greater context. The discussion need not be extensive, but MZ cells should be mentioned. In particular, IgM secreting cells mostly derive from MZ B cells. Furthermore, there are reports in the literature suggesting that MZ B cells are more likely to respond to self-antigens than other B cells. In addition, early on, the authors bring up the subject of nIgM, and state that the source of these antibodies are B1 B cells. However, MZ cells are also known to be a source of nIgM. In fact, there are multiple reports in the literature, which connect decreased expression of MZ cells to autoimmunity, which would fit in nicely with the authors’ discussion that decreased nIgM expression is associated with autoimmunity.
Finally, throughout the paper the authors talk about antigen dependent IgM mediated signals. In reality, most of the responding cells they are talking about likely also express IgD. Technically, antigen induced signals in these cells are a combination of membrane IgM and membrane IgD dependent signaling.
Author Response
#1 The subject of the connection of secreted IgM to autoimmunity is timely, and certainly of importance. This review is well written, thought provoking, and includes a current and comprehensive reference list. The discussion of the connection of COVID 19 to IgM dependent autoimmune phenomena is especially well timed.
A#1 We thank the reviewer for her/his appreciation of our manuscript and have modified the revised version according to the suggestions (see below). Changes in the manuscript are highlighted in pink.
#2 What is lacking though is some discussion of marginal zone (MZ) B cells, which would bring the whole discussion of secreted IgM into greater context. The discussion need not be extensive, but MZ cells should be mentioned. In particular, IgM secreting cells mostly derive from MZ B cells. Furthermore, there are reports in the literature suggesting that MZ B cells are more likely to respond to self-antigens than other B cells. In addition, early on, the authors bring up the subject of nIgM, and state that the source of these antibodies are B1 B cells. However, MZ cells are also known to be a source of nIgM. In fact, there are multiple reports in the literature, which connect decreased expression of MZ cells to autoimmunity, which would fit in nicely with the authors’ discussion that decreased nIgM expression is associated with autoimmunity.
A#2 We have included new paragraphs discussing MZ B cells and nIgM in the revised version of the manuscript (section 2).
#3 Finally, throughout the paper the authors talk about antigen dependent IgM mediated signals. In reality, most of the responding cells they are talking about likely also express IgD. Technically, antigen induced signals in these cells are a combination of membrane IgM and membrane IgD dependent signaling.
A#3 This is a very interesting aspect. Autoreactive B cells expressing the IgD-class BCR maintain unresponsiveness by being ignorant to monovalent (soluble) antigen. However, they can be activated when encountering polyvalent (complex) autoantigen. If this process happens to be the case in the described diseases is unknown. Since we have recently published a detailed review on B cell tolerance and the role of the IgD-class BCR in tolerance (Amendt & Jumaa 2022, BioEssays), we wanted to explicitly focus on IgM autoantibodies causing human diseases and the potential contribution of TLR7 in the current manuscript.
Nevertheless, since this is an important aspect, we have added a sentence briefly describing the current view.
Taken together, we thank reviewer#1 for her/his comments on our manuscript which improved its quality significantly.
Reviewer 2 Report
1) This is a very well-written review of IgM's role in autoimmunity. Limited literature available on IgM, this review would be a great addition
2) Authors did not mention IgM glycosylation while discussing the structure line 54-55. The role of IgG glycosylation is well established in maintaining immune homeostasis and the role of each glycan is characterized. Similarly, IgM glycosylation plays an important role in Immunomodulatory Effects. The figures also suggested a glycosylation site on IgG & IgM. It would be a good addition to the introduction by disusing and citing the following papers.
https://www.ncbi.nlm.nih.gov/pmc/articles/PMC9021442/
https://www.jimmunol.org/content/194/1/151
https://www.jimmunol.org/content/180/3/1780.long
3) Figures did not show a glycosylation site. Authors could put an asterisk and include it in the figure legend. As literature emerges it would be better not to exclude glycosylation.
4) Figure legends are very short. Elaborate figure legends
Author Response
1) This is a very well-written review of IgM's role in autoimmunity. Limited literature available on IgM, this review would be a great addition
We thank the reviewer for her/his appreciation of our manuscript and have modified the revised version according to the suggestions (see below). Changes in the manuscript are highlighted in green.
2) Authors did not mention IgM glycosylation while discussing the structure line 54-55. The role of IgG glycosylation is well established in maintaining immune homeostasis and the role of each glycan is characterized. Similarly, IgM glycosylation plays an important role in Immunomodulatory Effects. The figures also suggested a glycosylation site on IgG & IgM. It would be a good addition to the introduction by disusing and citing the following papers.
https://www.ncbi.nlm.nih.gov/pmc/articles/PMC9021442/
https://www.jimmunol.org/content/194/1/151
https://www.jimmunol.org/content/180/3/1780.long
We included a paragraph discussing the importance of antibody glycosylation within the introduction and modified figure 1 by adding glycosylation sites (marked with a blue triangle, Fig. 1). We believe that the addition of the mentioned discussion significantly improves the quality of the manuscript.
3) Figures did not show a glycosylation site. Authors could put an asterisk and include it in the figure legend. As literature emerges it would be better not to exclude glycosylation.
We have added N-linked glycosylation sites of antibodies by a triangle within the figure and referred to that in the improved figure legends.
4) Figure legends are very short. Elaborate figure legends
We have added more elaborated figure legends to enhance understanding of the figures (Fig. 1 and Fig. 2).
Taken together, we thank reviewer#2 for her/his comments on our manuscript which improved its quality significantly.
Reviewer 3 Report
This is a well written review that focuses on recent findings regarding the role of autoreactive IgM in autoimmune diseases. I have only minor comments.
-Authors have appropriately cited the risk of autoimmunity in patients with hyper IgM syndrome (HIGM). Since the link between primary or secondary HIGM and autoimmunity reinforces the role of IgM in the induction of autoimmunity, I suggest to focus more on this aspect. Some papers, including Barbouche et al- Cellular and Molecular Immunology, 2018; Gallo V et al - J Clin Med 2020) should be quoted.
- Paradoxically, autoimmunity is also reported in patients with IgM deficiency. Authors should comment these data.
Author Response
#1 This is a well written review that focuses on recent findings regarding the role of autoreactive IgM in autoimmune diseases. I have only minor comments.
A#1 We thank the reviewer for her/his appreciation of our manuscript and have modified the revised version according to the suggestions (see below). Changes in the manuscript are highlighted in blue.
#2 Authors have appropriately cited the risk of autoimmunity in patients with hyper IgM syndrome (HIGM). Since the link between primary or secondary HIGM and autoimmunity reinforces the role of IgM in the induction of autoimmunity, I suggest to focus more on this aspect. Some papers, including Barbouche et al- Cellular and Molecular Immunology, 2018; Gallo V et al - J Clin Med 2020) should be quoted.
A#2 We have added a paragraph about HIGM and IgM-mediated autoimmunity discussing possible mechanisms within section 5. As suggested by the reviewer, autoantibodies present in HIGM patients inducing autoimmune diseases support the concept of IgM as a critical player in autoimmune diseases.
#3 Paradoxically, autoimmunity is also reported in patients with IgM deficiency. Authors should comment these data.
A#3 This is a very interesting question. In the current manuscript we describe the pathogenic role of autoreactive IgM (primary IgM) targeting self-molecules and thereby leading to autoimmune diseases. However, the vast majority of serum IgM belongs to the class of natural IgM (nIgM). The so-called nIgM is known to possess strong polyreactivity and low antigen-affinity targeting altered and harmful self-molecules such as oxidized lipids, advanced glycation end-products (AGE) etc. In the case of IgM deficiency, all protective functions of nIgM are missing and mice/patients suffer from chronic inflammation and eventually develop autoimmune diseases such as lupus. In consequence, the reason why patients suffering from selective IgM deficiency develop autoimmune diseases is most likely the lack of nIgM which is crucial for homeostasis.
To make this point more clear, we included a paragraph about human sIgMD within the natural IgM section in the revised version of the manuscript. Importantly, nIgM targeting harmful altered self-molecules prevent chronic inflammation thereby suppressing autoimmune diseases, whereas IgM autoantibodies targeting receptors (self-molecules) induce autoimmune diseases themselves.
Taken together, we thank reviewer#3 for her/his comments on our manuscript which improved its quality significantly.